# Determinants of Long Working Hours Among Obstetrics and Gynecology Nurses and Midwives in Japan: A National Cross-Sectional Study

**DOI:** 10.3390/healthcare13121413

**Published:** 2025-06-12

**Authors:** Masatoshi Ishikawa, Ryoma Seto, Michiko Oguro, Yoshino Sato, Mayo Ogawa, Izumi Katagiri, Mini Kaneko

**Affiliations:** Tokyo Healthcare University, Tokyo 140-0022, Japan

**Keywords:** nurse, midwife, working hour, Japan, work reform

## Abstract

**Background/Objectives:** Nursing staff face mentally and physically demanding work environments in the obstetrics and gynecology departments in hospitals. This study elucidated the working hours of midwives and nurses in these departments and the background factors influencing them. **Methods:** This study employed a quantitative, descriptive, and correlational cross-sectional design. A questionnaire-based survey targeting nursing personnel working in the obstetrics and gynecology departments in hospitals across Japan was conducted. The respondents’ attributes, working hours, number of night shifts, and other employment conditions were described. To identify the background factors of long working hours, multivariate logistic regression analysis was performed using working hours ≥50 h per week as dependent variables and respondents’ attributes and employment conditions as explanatory variables. **Results:** Questionnaires were sent to 1170 hospitals, and valid responses were obtained from 2043 nursing personnel in 474 hospitals. Working ≥50 and ≥60 h per week were observed in 15.5% and 3.6% of the respondents, respectively, and 54.2% reported working night shifts ≥5 times monthly. Background factors strongly correlated with working ≥50 h per week among nursing staff included being in their 40s, licensed practical nurses, or a head nurse; having 5–8 night shifts per month; and working in hospitals with a total bed count of 200–400, 400–600, or 600–800, as well as ≥10 full-time physicians, ≥10 or an unknown number of advanced practice midwives, and >400 inpatient midwifery delivery cases annually. **Conclusions:** Urgent interventions are needed to reduce the workload of nursing staff in the obstetrics and gynecology departments of Japanese hospitals.

## 1. Introduction

The working hours of workers in Japan are longer compared with international standards [1]. Evidence concerning the negative health impacts of extended working hours is abundant. Indeed, a joint study by the World Health Organization (WHO) and the International Labor Organization (ILO) found that working ≥55 h per week increases the risk of ischemic heart disease by 17% and stroke by 35%, underlining the global burden of excessive work time [2,3]. A systematic review by Bannai et al. found an association between long working hours and conditions, such as depression, anxiety, sleep disturbances, and coronary heart disease [4]. When compared across professions, nursing staff tend to have relatively longer working hours. Among the medical departments, obstetrics and gynecology are noted for their longer working hours, although their specifics remain unclear [5].

Burnout is defined as a psychological syndrome of emotional exhaustion, depersonalization, and reduced personal accomplishment due to chronic occupational stress. In the nursing context, burnout includes physical fatigue and mental overload. The recent literature indicates that nurses, particularly in obstetric settings, face heightened burnout risks due to workload intensity and shift irregularities [6].

Nursing staff working in hospitals face mentally and physically demanding conditions that impact the quality and safety of healthcare [6]. Suicides caused by depressive symptoms related to overwork and deaths from ischemic heart disease and cerebrovascular diseases are among the conditions associated with “overwork death”, or “karoshi”, a significant public health concern especially observed in East Asian countries, such as Japan [7]. The Ministry of Health, Labor, and Welfare of Japan is currently spearheading the “Workstyle Reform for Physicians”. Workstyle reform for nursing staff is also a critical issue that demands attention [8].

Although the Labor Standards Act prescribes a 40 h workweek in Japan, overtime is commonly practiced, particularly in healthcare. The Ministry of Health, Labor, and Welfare is implementing workstyle reforms for physicians, but similar regulatory attention has not been given to nurses, necessitating the urgency in addressing long working hours among nursing staff.

Few studies have been conducted on the working hours of nursing staff. The Japanese Nursing Association conducted a survey on the working conditions of hospital nursing staff and reported that only 0.8% of the staff worked >20 h of overtime per month, although detailed data have not been made public [9]. Additionally, the role of nurse managers and organizational structures that emphasize streamlining tasks unrelated to direct patient care and actively addressing overtime have been suggested to alleviate the burden of overtime on staff [10]. In the United States, efforts to limit overtime hours and consecutive work hours have reduced the working hours of nursing staff [11,12].

The aim of this study was to examine long working hours and underlying factors among nursing staff working in the obstetrics and gynecology department, which is particularly known for long working hours among Japanese workers. We hypothesized that long working hours among obstetrics/gynecology nursing staff are influenced by a combination of individual, hospital-level, and regional/geographic factors. This study considers factors beyond working hours, including individual, hospital, and regional factors, to provide insights into healthcare policy.

## 2. Materials and Methods

### 2.1. Participants

This study targeted 1170 hospitals specializing in obstetrics and gynecology, whose hospital names are publicly available through the Hospital Bed Function Reporting System [13]. On 1 September 2024, a web-based questionnaire survey was administered to the nursing personnel in charge of the obstetrics and gynecology departments of these hospitals. The deadline for the responses was 30 September 2024.

Questionnaires were sent to 1170 hospitals, and valid responses were obtained from 2043 nursing personnel in 474 hospitals at a response rate of 40.5%.

### 2.2. Measurement

The question item for working hours was as follows: “On average, how many total hours per week do you work, including overtime and night shifts?”.

First, the respondent attributes (sex, age, qualifications, job title, highest level of education, marital status, age of youngest child, working hours per week, number of night shifts per month, foundational entity of the hospital, total number of beds, regional characteristics, number of full-time doctors, number of full-time midwives, number of advanced practice midwives, shift system, annual delivery count, number of inpatient midwifery delivery cases, and number of outpatient midwifery delivery cases) are described below (Table 1).

Age was categorized into five groups: <30, 30s, 40s, 50s, and ≥60. Qualifications included midwives, registered nurses, licensed practical nurses, and advanced practice midwives, with multiple answers allowed. Job titles were categorized into four groups: staff, chief nurse, head nurse, and others. The highest level of education was categorized into five groups: specialist school, junior college, 4-year university, graduate school, and others. The number of children was categorized into five groups: none, 1, 2, 3, and ≥4. The youngest child was categorized into five groups: none, preschool, primary school, middle school, and high school or older. Working hours per week were classified into five categories: <40 h, 40–50 h, 50–60 h, 60–70 h, and >70 h. The number of night shifts per month was classified into four categories: 0 shift, 1–4 shifts, 5–8 shifts, and ≥9 shifts. Hospital establishment type was divided into four categories: public/government-run institutions, national university hospitals, private university hospitals, and other private institutions (excluding private universities). Hospital size, based on the total number of beds, was grouped into five categories: <200 beds, 200–400 beds, 400–600 beds, 600–800 beds, and >800 beds. Regional characteristics were classified into three types based on combining population size and density data across the 344 secondary medical care areas: urban, intermediate, and rural areas [14].

The number of full-time doctors was categorized into four groups: <5, 5–10, ≥10, and unknown. The number of full-time midwives was categorized into four groups: <10, 10–20, 20–30, ≥30, and unknown. The number of advanced practice midwives was categorized into four groups: <5, 5–10, ≥10, and unknown. The shift system was categorized into three types: two-shift, three-shift, and neither. The annual delivery count, inpatient midwifery delivery case count, and outpatient midwifery delivery case count were categorized into five groups: not performed, <200, 200–400, ≥400, and unknown.

Variables were selected based on the literature and expert consultation regarding factors influencing nursing labor patterns. The questionnaire was administered via a secure web platform to enable broad participation and ensure anonymity.

### 2.3. Statistical Analyses

To clarify the background factors of long working hours for the nursing staff, a multivariate logistic regression analysis was performed, with the presence or absence of working ≥50 h per week as the dependent variable (thresholds aligned with labor law) and the attributes of the responding nursing staff (sex, age, qualifications, job title, highest level of education, working hours per week, number of night shifts per month, foundational entity of hospital, total number of beds, regional characteristics, number of full-time doctors, number of full-time midwives, number of advanced practice nurses, shift system, annual delivery count, number of inpatient midwifery delivery cases, and number of outpatient midwifery delivery cases) as the explanatory variables. The threshold of 50 h per week was selected based on the labor administration practices in Japan, where 40 h per week is the statutory standard. Exceeding 50 h reflects sustained overtime and aligns with thresholds used in overwork-related health risk studies and policy guidelines (e.g., karoshi criteria). Additionally, hospital types were categorized as public, national university, private university, and private hospitals to account for institutional variations in management structure and labor demand.

Statistical analyses were considered significant at *p*-values < 0.05. STATA 17.0 was used for all statistical analyses.

### 2.4. Ethical Consideration

This study was approved by the Tokyo Healthcare University Research Ethics Committee for Human Studies (Approval Number: Kyo-023-30B). The purpose of this study and measures to ensure secure data management were stated on the first page of the questionnaire. We also explained to the potential participants that their involvement in the study was voluntary. The results were analyzed separately from personal information to allow for the anonymity and confidentiality of personal information.

## 3. Results

The questionnaire was sent to 1170 hospitals nationwide, and valid responses were obtained from 2043 nursing personnel from 474 hospitals (response rate, 40.5%). The respondents’ attributes (sex, age, qualifications, job title, highest level of education, working hours per week, number of night shifts per month, foundational entity of the hospital, total number of beds, regional characteristics, number of full-time doctors, number of full-time midwives, number of advanced practice nurses, annual delivery count, number of inpatient midwifery delivery cases, and number of outpatient midwifery delivery cases) are shown in Table 1.

The 1640 midwives who responded to the survey represented 7.1% of the 23,109 hospital-employed midwives reported in a survey conducted in 2022 by the Ministry of Health, Labor, and Welfare [15].

The most common working hours per week were 40–50 h per week (61.8%), followed by ≥50 h per week (15.5%), and ≥60 h per week (3.6%). Table 2 shows the results of the multivariate logistic regression analysis.

With weekly working hours of ≥50 as the dependent variable, the analysis showed significant associations with the following factors: for those in their 40s (reference: <30), the odds ratio (OR) was 1.65 (95% confidence interval [CI], 1.01–2.70; *p* = 0.04); for licensed practical nurses (reference: non-licensed practical nurses), the OR was 0.44 (95% CI, 0.20–0.98; *p* = 0.04); for head nurses (reference: staff), the OR was 4.37 (95% CI, 2.72–7.01; *p* < 0.01); for those with 5–8 night shifts per month (reference: 0 night shifts), the OR was 2.16 (95% CI, 1.29–3.60; *p* < 0.01); for hospitals with 200–400 beds (reference: <200 beds), the OR was 2.31 (95% CI, 1.19–4.50; *p* = 0.01); for hospitals with 400– 600 beds (reference: <200 beds), the OR was 2.66 (95% CI, 1.37–5.17; *p* < 0.01); for hospitals with 600–800 beds (reference: <200 beds), the OR was 2.28 (95% CI, 1.20–5.53; *p* = 0.02); for hospitals with ≥10 full-time physicians (reference: <5), the OR was 1.94 (95% CI, 1.16–3.26; *p* = 0.01); for hospitals with ≥10 advanced practice nurses (reference: <5), the OR was 0.57 (95% CI, 0.33–1.00; *p* = 0.05); for unknown number of advanced practice nurses (reference: <5), the OR was 0.65 (95% CI, 0.43–1.00; *p* = 0.05); and for hospitals with ≥400 inpatient midwifery deliveries (reference: none), the OR was 7.74 (95% CI, 2.29–26.13; *p* < 0.01). Although not statistically significant, the analysis showed a relationship between the age of the youngest child being a preschooler (reference: none), with an OR of 0.56 (95% CI, 0.31–1.01; *p* = 0.05). No significant relationships were observed among sex, highest level of education, marital status, number of children, foundational entity of the hospital, regional characteristics, number of full-time midwives, shift system, annual delivery count, or number of inpatient midwifery delivery cases.

## 4. Discussion

The proportion of nursing staff working >50 h per week was 15.5% of all respondents, revealing the excessive labor faced by nursing staff in the obstetrics and gynecology departments of Japanese hospitals. In a previous study conducted by the Japanese Nursing Association, only 0.8% of the respondents reported >20 h of overtime per month; however, the results of this study significantly exceeded that figure [9].

Situations where long working hours continue for several prolonged periods of psychological stress over several months are closely associated with the development of mental health conditions and cardiovascular diseases, referred to as “karoshi levels”. This level of work is recognized as the threshold for labor-related fatalities [16,17]. The “karoshi level” is defined as >100 h of overtime in a month, or an average of >80 h of overtime per month for 2–6 months. This standard has been set at the level at which the risk of overwork-related deaths and health disorders significantly increases, and efforts are being made across society to prevent this [18].

The results of this study suggest that 3.6% of the respondents with a weekly working time of ≥60 h (equivalent to >80 h of overtime per month) are in a work environment that exceeds the “karoshi threshold”.

Our analysis revealed that long working hours were associated with personal and organizational characteristics, such as mid-career status, managerial roles, frequent night shifts, large hospital size, and high delivery volume. These findings suggest systemic and individual-level factors jointly contribute to prolonged work time. No significant relationships were observed among sex, the highest level of education, marital status, number of children, a foundational entity of the hospital, regional characteristics, number of full-time midwives, shift systems, annual delivery count, or outpatient midwifery delivery cases.

As there have been no previous studies on the background factors of nurses with long working hours, discussing these factors is difficult. Several hypotheses have been proposed.

First, regarding personal attributes, such as being in their 40s, not being licensed practical nurses, being head nurses, and working 5–8 night shifts, it is hypothesized that head nurses often carry extensive managerial responsibilities, including shift planning and staff coordination, likely contributing to prolonged working hours. Introducing IT-based systems for automated shift scheduling and documentation has been proposed to reduce this burden [19]. Additionally, transitioning from traditional 8 h to 12 h shifts may improve operational efficiency and work–life balance without increasing absenteeism. These structural changes should be considered as part of broader efforts to optimize staffing and reduce overtime. Head nurses are generally in their 40s and are typically not licensed practical nurses (instead, midwives or registered nurses), which may also be related. There seems to be room to advance the transfer of management functions, such as automating shift creation using information technology tools [19].

Regarding the association between working 5–8 night shifts and long working hours, it is natural that the more night shifts there are, the longer the working hours. Night shifts typically involve working for >16 h, which brings challenges, such as nurses’ reduced self-monitoring ability after night shifts [20]. Shifts >12 h increase the risk of professional fatigue, leading to several dangers associated with fatigue [21]. However, there are also arguments suggesting that 12 h shifts, compared with traditional shift models, offer a clearer distinction between work and leisure, making staff management easier and thus not contributing to an increase in absenteeism for health reasons [22]. Although not the focus of this study, the structure of night shifts is also an important consideration.

Although not statistically significant, a negative association was observed between working >50 h/week and having the youngest child of preschool age. This suggests that when children are preschool-aged, mothers may need to devote more time to childcare, leading to a reduction in overtime hours.

Regarding hospital attributes, a significant positive correlation was found between the number of beds, doctors, and advanced practice midwives. A larger number of beds and doctors suggest that economies of scale improve human productivity, which may not necessarily lead to increased overtime. However, it can be hypothesized that as the scale of a hospital grows, managerial tasks, such as those handled by head nurses, may increase, which can be related to this trend.

Japan lacks nationally mandated nurse-to-patient ratios, and staff shortages in obstetrics and gynecology units may increase reliance on overtime to ensure adequate patient care. This shortage likely exacerbates long working hours among nursing staff.

The significant positive association observed with the number of inpatient midwifery cases can be attributed to the fact that inpatient midwifery requires a substantial amount of effort from the nursing staff, including midwives. This likely contributes to an increase in working hours.

In efforts to reduce working hours, studies have indicated that by utilizing structured clinical knowledge to streamline nursing documentation time, overtime hours can be significantly reduced by 50% [23]. This type of example should be adopted to improve the efficiency of nursing duties.

In the United States, limiting overtime hours and consecutive working hours reduces the long working hours of nursing staff [9,10]. Meanwhile, in Japan, it is also necessary to identify nursing staff who experience excessive labor and implement measures to reduce their working hours.

### Limitations of the Study

This study has some limitations.

First, because participation was voluntary, there is a possibility of selection bias. Nevertheless, we collected valid responses from 2043 participants, accounting for 7.1% of the total number of hospital-employed midwives (23,109 individuals) across 474 hospitals, yielding a response rate of 40.5%. Considering the response rate and sample size, we believe that the data are sufficiently representative. However, because this study focused solely on the obstetrics and gynecology departments, the findings may not be generalizable to other medical specialties with different workloads.

Second, the self-administered nature of the questionnaire might have introduced an information bias. For example, the participants’ reported weekly working hours may not be entirely accurate, as the study did not include direct time tracking. Additionally, to improve response rates, technical terms were not defined within the questionnaire, which might have varied the interpretations among respondents.

Furthermore, there might have been a selection bias if individuals experiencing the most severe work burdens could not participate due to time constraints. Also, given the cross-sectional nature of this study, causal relationships between factors and working hours could not be confirmed.

Third, although this study identified statistical associations between long working hours and potential influencing factors, causality could not be established. Unmeasured confounding variables may have influenced the results that are related. For example, it is unclear whether long working hours are the result of individual choices or hospital management decisions.

## 5. Conclusions

This national cross-sectional study is the first in Japan to examine working hours and related factors among nursing staff in the obstetrics and gynecology departments. The findings revealed that 15.5% of the respondents worked ≥50 h per week, and 3.6% worked ≥60 h, indicating the presence of prolonged working hours in this sector. Several factors, including age, job title, number of night shifts, hospital size, and volume of inpatient midwifery deliveries, were significantly associated with long working hours.

These results highlight the complexity of the factors contributing to overwork among obstetrics and gynecology nursing staff, including both individual- and institutional-level influences. Particularly, the workload burden on head nurses and staff in large hospitals with high delivery volumes requires urgent attention.

Given the potential health risks associated with long working hours and the growing need for healthcare workforce reforms, this study provides important insights for policymakers. Interventions to reduce working hours, such as optimizing shift schedules, redistributing management responsibilities, and utilizing digital tools, may be effective strategies. Further studies are required to explore the causal relationships and develop evidence-based policies that promote sustainable work environments for nursing professionals.

Although this study focused on nursing staff in the obstetrics and gynecology departments, future studies should include comparisons with other high-stress medical departments, such as emergency medicine and intensive care units. Such comparisons would help distinguish between the universal factors contributing to long working hours across medical fields and those specific to particular departments. Identifying common and department-specific drivers of overtime is crucial for developing targeted and effective interventions to reduce excessive working hours and improve occupational health in healthcare settings.

Future studies should investigate other high-stress departments, such as emergency medicine and intensive care, which may have distinct patterns of overwork. Department-specific insights will be vital for crafting effective, specialty-tailored policy responses.

Although regional characteristics were not directly measured, hospital size and delivery volume may reflect broader geographic and demographic differences. Future policies should consider these structural aspects when designing interventions.

## Figures and Tables

**Table 1 healthcare-13-01413-t001:** Characteristics of the participants.

	Total		≥50 h per Week	
Total number of participants, n	2043		317	
% of all participants	100.0%		15.5%	
Sex, n, %				
Female	2041	99.9%	316	99.7%
Male	2	0.1%	1	0.3%
Age, n, %				
<30	434	21.2%	50	15.8%
30–39	503	24.6%	64	20.2%
40–49	602	29.5%	106	33.4%
50–59	443	21.7%	93	29.3%
≥60	61	3.0%	4	1.3%
Qualifications				
Midwife	1640	80.3%	252	79.5%
Registered nurse	1463	71.6%	215	67.8%
Licensed practical nurse	99	4.8%	8	2.5%
Advanced practice midwife	397	19.4%	68	21.5%
Job title				
Staff	1462	71.6%	187	59.0%
Chief	360	17.6%	61	19.2%
Head nurse	180	8.8%	63	19.9%
Others	41	2.0%	6	1.9%
Highest level of education				
Specialist school	989	48.4%	162	51.1%
Junior college	353	17.3%	51	16.1%
4-year university	488	23.9%	70	22.1%
Graduate school	146	7.1%	26	8.2%
Others	67	3.3%	8	2.5%
Married?				
Yes	798	39.1%	119	37.5%
No	1245	60.9%	198	62.5%
No. of children				
None	904	44.2%	134	42.3%
1	241	11.8%	38	12.0%
2	543	26.6%	86	27.1%
3	305	14.9%	48	15.1%
≥4	50	2.4%	11	3.5%
Age of youngest child				
None	904	44.2%	134	42.3%
Preschool	306	15.0%	30	9.5%
Primary school	270	13.2%	45	14.2%
Middle school	112	5.5%	15	4.7%
High school or older	451	22.1%	93	29.3%
Working hour per week, n, %				
<40 h per week	463	22.7%	0	0.0%
40–50 h per week	1263	61.8%	0	0.0%
50–60 h per week	244	11.9%	244	77.0%
60–70 h per week	51	2.5%	51	16.1%
≥70 h per week	22	1.1%	22	6.9%
No. of night shifts per month				
0	275	13.5%	30	9.5%
1–4	660	32.3%	111	35.0%
5–8	940	46.0%	157	49.5%
≥9	168	8.2%	19	6.0%
Entity of employer				
Public	1262	61.8%	200	63.1%
National university	551	27.0%	76	24.0%
Private university	131	6.4%	28	8.8%
Private	99	4.8%	13	4.1%
Employer’s total no. of beds				
<200 beds	176	8.6%	14	4.4%
≥200–<400 beds	632	30.9%	90	28.4%
≥400–<600 beds	753	36.9%	131	41.3%
≥600–<800 beds	294	14.4%	54	17.0%
≥800 beds	188	9.2%	28	8.8%
Area, n, %				
Urban	693	33.9%	105	33.1%
Intermediate	1062	52.0%	173	54.6%
Rural	288	14.1%	39	12.3%
No. of full-time obstetricians				
<5	841	41.2%	108	34.1%
5–9	858	42.0%	143	45.1%
≥10	293	14.3%	59	18.6%
Unknown	51	2.5%	7	2.2%
No. of full-time midwives				
<10	389	19.0%	50	15.8%
10–20	763	37.3%	126	39.7%
20–30	450	22.0%	72	22.7%
≥30	384	18.8%	61	19.2%
Unknown	57	2.8%	8	2.5%
No. of full-time advanced practice midwives				
<5	803	39.3%	133	42.0%
5–9	564	27.6%	108	34.1%
≥10	222	10.9%	28	8.8%
Unknown	454	22.2%	48	15.1%
Shift system				
Two-shift	1461	71.5%	232	73.2%
Three-shift	482	23.6%	68	21.5%
Neither	100	4.9%	17	5.4%
Annual delivery count				
None	45	2.2%	2	0.6%
<200	509	24.9%	75	23.7%
200–400	714	34.9%	115	36.3%
≥400	642	31.4%	109	34.4%
Unknown	133	6.5%	16	5.0%
No. of inpatient midwifery cases per year				
None	1468	71.9%	216	68.1%
<200	302	14.8%	54	17.0%
200–400	43	2.1%	9	2.8%
≥400	13	0.6%	7	2.2%
Unknown	217	10.6%	31	9.8%
No. of outpatient midwifery cases per year				
None	572	28.0%	75	23.7%
<200	733	35.9%	109	34.4%
200–400	253	12.4%	51	16.1%
≥400	274	13.4%	53	16.7%
Unknown	211	10.3%	29	9.1%

**Table 2 healthcare-13-01413-t002:** Logistic regression analysis of factors associated with longer working hours.

>50 h per Week			
	OR	95% CI	*p*-Value
Sex
Female	Reference
Male	10.32	0.55–192.45	0.12
Age
<30	Reference
30s	1.33	0.85–2.07	0.21
40s	1.65	1.01–2.70	0.04
50s	1.32	0.76–2.30	0.33
≥60	0.46	0.14–1.55	0.21
Qualifications
Midwife	0.97	0.64–1.47	0.89
Registered nurse	0.89	0.66–1.20	0.44
Licensed practical nurse	0.44	0.20–0.98	0.04
Advanced practice midwife	0.89	0.62–1.27	0.52
Job title
Staff	Reference
Chief	1.15	0.80–1.66	0.46
Head nurse	4.37	2.72–7.01	<0.01
Others	1.23	0.47–3.18	0.67
Highest level of education
Specialist school	Reference
Junior college	0.82	0.57–1.20	0.31
4-year university	0.87	0.62–1.24	0.45
Graduate school	0.95	0.57–1.58	0.84
Others	0.69	0.31–1.51	0.35
Married?
No	Reference
Yes	1.05	0.73–1.51	0.79
No. of children
None	Reference
1	1.22	0.70–2.13	0.49
2	1.08	0.65–1.80	0.76
3	1.12	0.63–2.00	0.69
4 or more	2.56	1.06–6.17	0.04
Age of the youngest child
None	Reference
Preschool	0.56	0.31–1.01	0.05
Primary school	0.72	0.44–1.19	0.20
Middle school	0.55	0.29–1.06	0.07
High school or older	1.00		
No. of night shifts per month

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

## Data Availability

Data available on request due to privacy and ethical reasons.

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
