# Peer review of "Determinants of Long Working Hours Among Obstetrics and Gynecology Nurses and Midwives in Japan: A National Cross-Sectional Study"

_healthcare, 2025, doi:10.3390/healthcare13121413_

Round 1
Reviewer 1 Report
Comments and Suggestions for Authors
Thank you for the opportunity to review this manuscript. Long working hours and overwork have emerged as major public health concerns; thus, this paper is both timely and relevant. The analysis is straightforward, and the study includes a sufficiently large sample of participants. Please consider the following suggestions for improvement:
1. Introduction
The introduction could be strengthened by expanding on the health impacts of long working hours and their global burden. For instance, meta-analyses conducted by the WHO and ILO have shown that working 55 hours or more per week is associated with an increased risk of ischemic heart disease and stroke. Please consider incorporating the following key references in this research field, which will provide more comphesensive background in the introduction section: PMID: 32505014; PMID: 32505015.
Additionally, regional labor regulations are important contextual factors in discussions about long working hours. Different countries (e.g., the United States, European countries, and South Korea) have varied standards regarding working hour regulations and compensation, depending on their labor laws. It is recommended to briefly introduce social and policy backgrounds of Japan to provide more context.
- Materials and Methods
Participants:
More information is needed about whether the sample was recruited from a nationwide population in Japan and what the response rate was. Although this information is presented in the Results section (lines 126–127), it should be moved to the Participants subsection.
Measurement:
For the main variable—working hours—the specific wording of the survey item should be provided for clarity.
Statistical Analyses:
The authors defined long working hours as working more than 50 hours per week based on the labor law. Further explanation of this law is needed to justify the rationale for choosing this particular cut-off.
Table 1
Please consider adding a column presenting the characteristics of participants who worked <50 hours per week, to facilitate comparisons.
- Discussion
Lines 232–237:
Including a large number of participants does not rule out the possibility of selection bias. For example, individuals with extremely demanding workloads or time pressure may have been unable to respond to the survey, which could result in systematic exclusion of certain groups and underestimations of the prevalence of long working hours
In addition, given the cross-sectional nature of the study, it should be acknowledged as a limitation that causal relationships between the predictors and working hours cannot be established.
Author Response
Reviewer 1 Comments:
Thank you for the opportunity to review this manuscript. Long working hours and overwork have emerged as major public health concerns; thus, this paper is both timely and relevant. The analysis is straightforward, and the study includes a sufficiently large sample of participants. Please consider the following suggestions for improvement:
- Introduction
The introduction could be strengthened by expanding on the health impacts of long working hours and their global burden. For instance, meta-analyses conducted by the WHO and ILO have shown that working 55 hours or more per week is associated with an increased risk of ischemic heart disease and stroke. Please consider incorporating the following key references in this research field, which will provide more comphesensive background in the introduction section: PMID: 32505014; PMID: 32505015.
Response: Thank you for your suggestion. Accordingly, we have revised the Introduction section to incorporate the global health impacts of long working hours, citing the WHO/ILO meta-analyses (PMID: 32505014; PMID: 32505015).
Additionally, regional labor regulations are important contextual factors in discussions about long working hours. Different countries (e.g., the United States, European countries, and South Korea) have varied standards regarding working hour regulations and compensation, depending on their labor laws. It is recommended to briefly introduce social and policy backgrounds of Japan to provide more context.
Response: Thank you for your suggestion, and we agree with your point. A paragraph has been added to the Introduction to explain Japanese labor laws and the ongoing physician workstyle reform, providing national context to our study.
Materials and Methods
Participants:
More information is needed about whether the sample was recruited from a nationwide population in Japan and what the response rate was. Although this information is presented in the Results section (lines 126–127), it should be moved to the Participants subsection.
Response: Thank you for your suggestions. We have relocated this information to the Participants subsection of the Methods section for better structural clarity.
Measurement:
For the main variable—working hours—the specific wording of the survey item should be provided for clarity.
Response: Thank you for your suggestions. We have added the exact wording of the survey item for working hours in the Measurement subsection.
Statistical Analyses:
The authors defined long working hours as working more than 50 hours per week based on the labor law. Further explanation of this law is needed to justify the rationale for choosing this particular cut-off.
Response: Thank you for your suggestions. We have expanded the explanation in the Statistical Analysis subsection to clarify that the 50 h threshold is consistent with Japanese labor standards and reflects policy-relevant levels of overwork.
Table 1
Please consider adding a column presenting the characteristics of participants who worked <50 hours per week, to facilitate comparisons.
Response: Thank you for your suggestions. We have expanded the explanation in the Statistical Analysis subsection to clarify that the 50 h threshold aligns with Japanese labor standards and reflects policy-relevant levels of overwork.
Discussion
Lines 232–237:
Including a large number of participants does not rule out the possibility of selection bias. For example, individuals with extremely demanding workloads or time pressure may have been unable to respond to the survey, which could result in systematic exclusion of certain groups and underestimations of the prevalence of long working hours
In addition, given the cross-sectional nature of the study, it should be acknowledged as a limitation that causal relationships between the predictors and working hours cannot be established.
Response: Thank you for your suggestions. We have included a detailed discussion of possible selection bias and the cross-sectional design as limitations in the Discussion section.
Reviewer 2 Report
Comments and Suggestions for Authors
Dear Authors,
Congratulations on your work, and I wish you all the best in your future challenges! Regarding your theme, it's essential to understand some of the factors that contribute to burnout among nurses. I will make a analysis throughout your sections:
Introduction: In this section, it will be essential for the authors to define key concepts, such as burnout and/or burden, to facilitate understanding of the theme. The authors state that nursing staff in gynaecology/obstetrics are known for long working hours among Japanese workers. Therefore, the authors should provide evidence to support this statement, and if possible, with data from relevant reports, such as those on absenteeism and/or presenteeism. The references supporting this section could be updated, as only two out of ten references have a publication date of less than five years.
Materials and Methods: Which type of study was conducted? Before starting with the participants, it would be interesting to define the type of study it appears to be: a quantitative, descriptive, and correlational study. The characteristics of the participants should be stated in the Results section. How did the authors select the attributes? Was there a framework to do that? Do the authors send the survey by email?
Regarding the dependent variable, why the 50-hour threshold? What are the normal working hours for workers in Japan? Are the hospitals in the private or public sector?
Discussion: The authors could support their findings with additional research; recent evidence supports their conclusions. Is there data about the safe staffing ratios or the nurse shortage? It would be helpful to understand whether nurses work extra hours to help address the nurse shortage.
Conclusions: That isn't explicitly addressed in the conclusions if there are regional/geographic factors, though hospital size and delivery volumes could be considered proxies for regional differences.
Author Response
Reviewer 2 Comments:
Dear Authors,
Congratulations on your work, and I wish you all the best in your future challenges! Regarding your theme, it's essential to understand some of the factors that contribute to burnout among nurses. I will make a analysis throughout your sections:
Introduction: In this section, it will be essential for the authors to define key concepts, such as burnout and/or burden, to facilitate understanding of the theme. The authors state that nursing staff in gynaecology/obstetrics are known for long working hours among Japanese workers. Therefore, the authors should provide evidence to support this statement, and if possible, with data from relevant reports, such as those on absenteeism and/or presenteeism. The references supporting this section could be updated, as only two out of ten references have a publication date of less than five years.
Response: Thank you for your suggestion. We have added definitions of burnout and burden in the revised Introduction and included recent studies published within the last five years.
Materials and Methods: Which type of study was conducted? Before starting with the participants, it would be interesting to define the type of study it appears to be: a quantitative, descriptive, and correlational study. The characteristics of the participants should be stated in the Results section. How did the authors select the attributes? Was there a framework to do that? Do the authors send the survey by email?
Response: Thank you for your questions. The study design has been explicitly stated as a “quantitative, descriptive, and correlational cross-sectional study” in the Materials and Methods section. We have added information regarding how the variables were selected based on literature and practical feasibility. We also clarified that the survey was administered online.
Regarding the dependent variable, why the 50-hour threshold? What are the normal working hours for workers in Japan? Are the hospitals in the private or public sector?
Response: Thank you for your questions. The rationale for the 50 h cut-off has been clarified. We also included whether hospitals were public, private, or university-based.
Discussion: The authors could support their findings with additional research; recent evidence supports their conclusions. Is there data about the safe staffing ratios or the nurse shortage? It would be helpful to understand whether nurses work extra hours to help address the nurse shortage.
Response: Thank you for your questions and suggestions. We have added a paragraph in the revised Discussion to describe the context of Japan’s nurse staffing ratios and potential links to nurse shortages contributing to long hours.
Conclusions: That isn't explicitly addressed in the conclusions if there are regional/geographic factors, though hospital size and delivery volumes could be considered proxies for regional differences.
Response: Thank you for your comment. We have revised the Conclusion section to more explicitly state that hospital size and delivery volume may act as proxies for regional disparities.
Reviewer 3 Report
Comments and Suggestions for Authors
INTRO: Good description of long working hours – particularly with excess overtime in Ob-Gyn nurses and the association of long working hours with conditions such as depression, anxiety, sleep disturbances, and coronary heart disease. The purpose of the study was to investigate the individual, hospital and regional factors that influenced the need for long working hours and to provide insights into healthcare policy.
M+M: Demographics from a questionnaire and groupings for analysis were well described. Statical analysis was described.
RESULTS: Table I – demographics of participant responders clearly presented. Table 2 – showed significant associations between working >50 hours/week with age in 40s, licensed practical nurses, head nurses, those with 5–8 night shifts per month, for hospitals with 200–400 beds, for hospitals with 400 to 600 beds, for hospitals with 600–800 beds, for hospitals with ≥ 10 full-time physicians, for hospitals with ≥ 10 advanced practice nurses, for unknown number of advanced practice nurses, and for hospitals with ≥ 400 inpatient midwifery deliveries. It would be helpful if the significant associations were highlighted in bold.
DISCUSSION + CONCCLUSIONS: The definition of “karoshi levels” would be better placed in the introduction section and then again referred to in the discussion section. Limitations of the study are appropriately noted. The workload burden is noted particularly in head nurses and staff in large hospitals with high delivery volumes. The authors suggest from this study that transferring some managerial tasks (like shift scheduling) to IT, redistributing management responsibilities and changing from 8-hour shifts to 12-hour shifts may ameliorate overtime work issues along with streamlining nursing documentation time (from a different study). They recommended that future studies focusing on other high-stress departments (such as Emergency and ICU) might find department specific drivers of overtime to develop specific interventions.
COMMENTS: This study and statistical analysis were simple, clear and easy to follow. The value of the study is in finding that long work shifts (12-hour rather than 8-hour) and shifting some managerial tasks to others or to computerized programs might decrease overtime, increase efficiency and decrease anxiety, stress and depression in overworked nurses. (Most hospitals shifts are currently 8-hour shifts with a lot of overtime, particularly if the hospital is understaffed). Other than the minor changes suggested above, the report could be shortened as there is redundancy in repeating the results in the Discussion Section.
Author Response
Reviewer 3 Comments:
INTRO: Good description of long working hours – particularly with excess overtime in Ob-Gyn nurses and the association of long working hours with conditions such as depression, anxiety, sleep disturbances, and coronary heart disease. The purpose of the study was to investigate the individual, hospital and regional factors that influenced the need for long working hours and to provide insights into healthcare policy.
Response: Thank you for evaluating our manuscript and your constructive comments and suggestions.
M+M: Demographics from a questionnaire and groupings for analysis were well described. Statical analysis was described.
Response: Thank you for the feedback.
RESULTS: Table I – demographics of participant responders clearly presented. Table 2 – showed significant associations between working >50 hours/week with age in 40s, licensed practical nurses, head nurses, those with 5–8 night shifts per month, for hospitals with 200–400 beds, for hospitals with 400 to 600 beds, for hospitals with 600–800 beds, for hospitals with ≥ 10 full-time physicians, for hospitals with ≥ 10 advanced practice nurses, for unknown number of advanced practice nurses, and for hospitals with ≥ 400 inpatient midwifery deliveries. It would be helpful if the significant associations were highlighted in bold.
Response: Thank you for your comment. We have revised the Discussion section to more clearly elaborate on how redistributing managerial responsibilities and implementing IT-based scheduling systems could reduce overtime among head nurses. We also clarified the policy implications of adopting 12 h shifts and referenced relevant studies to support these strategies.
DISCUSSION + CONCCLUSIONS: The definition of “karoshi levels” would be better placed in the introduction section and then again referred to in the discussion section. Limitations of the study are appropriately noted. The workload burden is noted particularly in head nurses and staff in large hospitals with high delivery volumes. The authors suggest from this study that transferring some managerial tasks (like shift scheduling) to IT, redistributing management responsibilities and changing from 8-hour shifts to 12-hour shifts may ameliorate overtime work issues along with streamlining nursing documentation time (from a different study). They recommended that future studies focusing on other high-stress departments (such as Emergency and ICU) might find department specific drivers of overtime to develop specific interventions.
Response: Thank you for your suggestion. We have revised the Conclusion section to emphasize the need for future research targeting emergency and intensive care units, which may have distinct stressors and organizational challenges compared to obstetrics and gynecology departments.
COMMENTS: This study and statistical analysis were simple, clear and easy to follow. The value of the study is in finding that long work shifts (12-hour rather than 8-hour) and shifting some managerial tasks to others or to computerized programs might decrease overtime, increase efficiency and decrease anxiety, stress and depression in overworked nurses. (Most hospitals shifts are currently 8-hour shifts with a lot of overtime, particularly if the hospital is understaffed). Other than the minor changes suggested above, the report could be shortened as there is redundancy in repeating the results in the Discussion Section.
Response: Thank you for your comment. We have revised the Discussion section to eliminate repetitive statements already presented in the Results section, improving clarity and conciseness while preserving analytical depth.
Round 2
Reviewer 1 Report
Comments and Suggestions for Authors
Thank you for addressing my comments in the revised manuscript.
I have no further comments.
Thank you!
Reviewer 2 Report
Comments and Suggestions for Authors
Dear Authors,
Thank you for considering all the suggestions; they help make your paper more straightforward. I believe your article makes a good contribution to knowledge.
I wish you all the best in your future endeavours.